# Annulation of Perimidines with 5-Alkynylpyrimidines en Route to 7-Formyl-1,3-Diazopyrenes

**DOI:** 10.3390/ijms232415657

**Published:** 2022-12-10

**Authors:** Stanislav V. Shcherbakov, Alexander V. Aksenov, Maksim V. Vendin, Viktoria Yu. Shcherbakova, Anna Yu. Ivanova, Maksim O. Shcheglov, Sergei N. Ovcharov, Michael Rubin

**Affiliations:** 1Department of Chemistry, North Caucasus Federal University, 1a Pushkin St., 355017 Stavropol, Russia; 2Department of Chemistry, University of Kansas, 1567 Irving Hill Road, Lawrence, KS 66045, USA

**Keywords:** annulations, acetylenes, nitrogen heterocycles, Brønsted acid catalysis, rearrangements

## Abstract

Unusual rearrangements were shown to accompany Brønsted acid-assisted peri-annulations of 1*H*-perimidines with 5-alkynylpyrimidines. These transformations take different routes depending on the nature of acetylene precursor, and lead to the formation of 7-formyl-1,3-diazopyrenes.

## 1. Introduction

Due to their unique physicochemical properties, pyrene derivatives have emerged as some of the most privileged structures in the design of organic fluorescent materials. Pyrene motifs attract attention of many research groups studying photochemistry and molecular electronics. Examples demonstrating utilization of pyrenes in the manufacturing of organic light-emitting diodes (OLED) [1,2,3], organic filed-effect transistors (OFET) [2,4,5,6], organic photo-voltaic devises (OPVs) [2,7,8], hole-conductive materials for solar cells [9], and other photoconductive covalent organic building blocks are omnipresent in literature [10]. Major advances were made in the development of pyrene-based fluorescent probes [11] for the analytical detection of copper [12,13] and other heavy metals [13], as well as picric acid [14]. These versatile synthons also possess a great intercalating ability to selectively bind to DNA in cellular nuclei [15,16]. The undisputed advantages of pyrene derivatives are outweighed by one significant drawback–their low solubility in common organic solvents—which complicates synthesis of advanced synthetic precursors for the manufacturing of photosensors and electronic devices. Another problem is associated with high carcinogenicity of these compounds and their slow metabolism, which severely limits their application in medicinal and pharmaceutical chemistry. Both issues could be addressed by incorporation of nitrogen atoms in the pyrene structure, simultaneously providing a powerful tool for the fine-tuning of photochemical and electrochemical properties of the resulting products. Our research group has a pioneering expertise in the development of synthetic methods for peri-annulation of carbo- and azacyclic compounds [17,18,19,20]. 1*H*-Perimidines **1** are typically employed as model substrates in these investigations, since they are characterized by increased electron density in the peri-position, making them excellent nucleophilic synthons. Reactions of 1*H*-perimidines with chalcones **2** [21] and pyrimidines **4** [19], which proceed in acidic media and afford derivatives of 1,3-diazapyrenes **3**, deserve a special note as an expeditious one-step route to 1,3-diazopyrenes (Figure 1). Herein, we disclose an alternative approach to 1,3-diazopyrenes **6** or **7** via annulation of 1*H*-perimidines **1** with 5-alkynylpyrimidines **5** (Figure 1). This reaction allows for selective installation of the formyl group at C-7 amenable for further synthetic modifications.

## 2. Results and Discussion

Following our earlier work on peri-annulation, we have recently reported a novel method for preparation of 7*H*-imidazo[4′,5′:4,5]benzo[1,2,3-*gh*]perimidines **9** via a reaction of perimidines **1** with 5-bromopyrimidines **8** in polyphosphoric acid (PPA) (Figure 2). Similar results were obtained upon heating in PPA of intermediate **10**, which, in turn, could be obtained by arylation of perimidines **1** with bromopyrimidines **8** in methanesulfonic acid at ambient temperature (Figure 2). To further advance this methodology, we attempted the reaction of **1** with 5-alkynylpyrimidines **5** in methanesulfonic acid aiming at dihydroquinazolino[6,7,8-*gh*]perimidines **15** via the following sequence (Figure 3). 

It was proposed that an S*_E_*Ar-type reaction of perimidine **1** would take place, in which the protonated form of pyrimidine **5** would act as an electrophile. The reaction would occur at the electron-rich peri*-*position in **1** to form sigma-complex **11**, which, after re-aromatization, would produce 7-(3,4-dihydropyrimidin-4-yl)-1*H*-perimidine **12** (Figure 3). The latter should be well-set for the subsequent nucleophilic 6-*exo-dig* cyclization, leading to 6-alkylidene-5a,6,10,10a-tetrahydroquinazolino[6,7,8-*gh*]perimidin-1-ium species **13**, which would further re-aromatize into 1,6,10,10a-tetrahydroquinazolino[6,7,8-*gh*]perimidine **14** (Figure 3). Finally, the pyrimidine moiety in **14** would undergo an ANRORC cascade via a ring opening and the subsequent 6*-exo-trig* cyclization of exo-alkylidene moiety to afford pentacyclic product **15**.

To evaluate this idea, we carried out a reaction between 2-phenyl-1*H*-perimidine (**1a**) with 5-(hept-1-yn-1-yl)pyrimidine (**5a**) in methanesulfonic acid at room temperature. Contrary to our expectations, the reaction did not afford product **15**. Instead, 1,3-diazapyrene **16a** possessing a *n*-hexyl substituent at C-6 and an aldehyde moiety at C-7 was obtained as a sole isolable product in modest yield (Figure 4). The reaction proceeded to completion consuming both starting materials **1a** and **5a**, but a significant amount of product decomposed, as indicated by the formation of notable amounts of polymeric tars. The same outcome was observed in the reaction of alkyne **5a** with other perimidines (**1b**-**g**) affording a series of 7-formyl-1,3-diazapyrenes in low to moderate yield (Figure 4).

Next, we explored the possibility to perform this reaction with perimidine **1a** using 5-(phenylethynyl)pyrimidine (**5b**) as the electrophilic component. Interestingly, the reaction took a different route leading to the formation of 1,3-diazopyrene **17a** bearing a benzylamine moiety at C-6 (Figure 4). Similarly to the example above, this reaction was rather general with respect to a variety of perimidines **1a**-**f**,**g** affording the corresponding 1,3-diazopyrenes **17a**-**f**,**g** as sole isolable products in moderate yield. The material balance in both of these reactions was far from perfect due to significant polymerization of the products. However, in all cases, the polymers were easily separated via a simple filtration through a short path silica gel column.

It was rationalized that formation of 7-formyl-1,3-diazapyrenes **16** and **17** may occur via two related cascade transformations depicted in Figure 5. The initially produced dihydroquinazolino[6,7,8-*gh*]perimidines **15** (Figure 3) are unstable under strongly acidic conditions, but their further reactivity is strongly dependent on the nature of the substituent at C-10a. *n*-Hexyl-substituted derivative **15a** undergoes electrocyclic cleavage of the dehydropyrimidine ring to establish aromaticity of 1,3-diazapyrene core (Figure 5). The resulting intermediate **18** bearing a masked aldehyde functionality in a form of acyclic formamidine moiety is highly susceptible to acidic hydrolysis. The removal of this protecting group should afford 7-formyl-1,3-diazapyrene **16** (Figure 5). Benzyl-substituted analog **15b** reacts via an alternative mechanistic pathway due to a much greater migratory aptitude of the benzyl group. Since **15b** has four nitrogen atoms with nearly identical basicity, it can form several protonated species coexisting in a dynamic equilibrium in the strongly acidic reaction medium. One of such forms (**19**) is an intermediate in tautomerization of 10,10a-dihydro-(**15**) into 1,10a-dihydroquinazolino[6,7,8-*gh*]perimidine **20** (Figure 5). Protonation of the latter triggers 1,2-migration of the benzyl moiety to N-10a to furnish **21**. A significant energy release accompanying aromatization of 1,3-diazopyrene serves as a strong driving force for this transformation. Furthermore, formation of a more basic *sp^3^*-hybridized nitrogen atom should be favored in an acidic medium. The non-aromatic heterocyclic ring in **21** is essentially a (methyleneamino)methanamine, which collapses under acidic hydrolysis conditions to furnish the benzylamine substituent at C-6 and an aldehyde group at C-7 in product **17**.

In support of this mechanistic rationale, we were able to isolate the intermediate 1,6,10,10a-tetrahydroquinazolino[6,7,8-*gh*]perimidines **14c,e** in low yield after quenching the corresponding reactions after 20 min (Figure 3). These stable crystalline materials were purified and re-subjected to the same reaction conditions to afford the expected products **17c**,**e** (Figure 6). The presence of 6-benzylamino-7-formyl-1,3-diazapyrene motif in product **17b** was confirmed by 2D NMR spectroscopy, including ^1^H-^1^H COSY, ^1^H-^13^C HSQC, ^1^H-^13^C HMBC, ^1^H-^15^N HSQC, and ^1^H-^15^N HMBCGP experiments (see Appendix A for details, Appendix A and Appendix A).

## 3. Methods and Materials


**General**


The NMR spectra, ^1^H, and ^13^C were measured in solutions of CDCl_3_ or DMSO-*d*_6_ on a Bruker AVANCE-III HD instrument (at 400.40 or 100.61 MHz, respectively). The residual solvent signals were used as internal standards in DMSO-*d*_6_ (2.50 ppm for ^1^H, and 40.45 ppm for ^13^C nuclei) or in CDCl_3_ (7.26 ppm for ^1^H, and 77.16 ppm for ^13^C nuclei). The high-resolution mass spectra were registered with a Bruker Maxis spectrometer (electrospray ionization, in MeCN solution, using HCO_2_Na–HCO_2_H for calibration). See Appendix A for NMR (Appendix A) and HRMS (Appendix A) spectral charts. The IR spectra were measured on FT-IR spectrometer Shimadzu IR Affinity-1S equipped with an ATR sampling module. The melting points were measured with a Stuart SMP30 apparatus. The reaction progress and purity of isolated compounds were controlled by TLC on ALUGRAM Xtra SIL G UV 254 plates. The column chromatography was performed with Macherey Nagel Silica gel 60 (particle size: 0.063–0.2 mm). The pyrimidines were synthesized by published methods [22,23] and the synthesis of 5-ethynylpyrimidines is described in our recent report [24]. All other reagents and solvents were purchased from commercial vendors and used as received.


**Method for preparation of benzo[*gh*]perimidine-7-carbaldehyde:**


A round bottom flask (10 mL) was charged with the appropriate pyrimidine **5** (1.00 mmol) and the corresponding 1*H*-perimidine **1** (1.00 mmol) was added, followed by 5 mL of methanesulfonic acid. The reaction mixture was stirred at room temperature and the reaction progress was monitored by thin-layer chromatography (EtOAc/Hexane, 2:1, *v*/*v*). After one hour, when TLC confirmed that all starting substances were consumed, the reaction mixture was poured isolated into cold water (50 mL) and neutralized with an aqueous ammonia solution (20%, 15 mL). The crystalline precipitate was filtered off and washed with a small amount of water to remove the excess ammonia. The resulting product was separated and purified by column chromatography (EtOAc/Hexane, 1:3, *v*/*v*).

*6-Hexyl-2-phenylbenzo[gh]perimidine-7-carbaldehyde* (**16a**): This compound was prepared by employing 2-phenyl-1*H*-perimidine **1a** and 5-(hept-1-yn-1-yl)pyrimidine **5a** in a yield of 137 mg (0.35 mmol, 35%). Purification was performed by column chromatography (EtOAc/Hexane = 1:3). The titled compound was obtained as light-yellow powder, m.p. 180.7–181.9 °C, R*_f_* 0.58 (EtOAc/Hexane, 1:3). ^1^H NMR (400 MHz, CDCl_3_) *δ* 10.76 (s, 1H), 8.88–8.79 (m, 4H), 8.53 (d, *J* = 9.1 Hz, 1H), 8.35 (d, *J* = 9.8 Hz, 1H), 8.25 (d, *J* = 9.1 Hz, 1H), 7.65–7.55 (m, 3H), 3.89–3.70 (m, 2H), 1.90–1.79 (m, 2H), 1.68–1.53 (m, 2H), 1.46–1.32 (m, 4H), 0.92 (t, *J* = 6.9 Hz, 3H); ^13^C{^1^H} NMR (101 MHz, DMSO-*d*_6_) *δ* 193.2, 161.7, 154.9, 154.1, 144.8, 138.1, 137.5, 136.5, 133.4, 131.1, 131.0, 128.8(2C), 128.6(2C), 127.9, 127.7, 127.4, 127.2, 124.6, 114.2, 33.4, 31.1, 28.9, 26.6, 22.1, 14.0; FTIR, *v_max_*: 3068, 2954, 1951, 1689, 1628, 1511, 1398, 1306, 909, 845, 794, 714, 695, 679, 668 cm^−1^; HRMS (ESI TOF) *m*/*z*: calc’d for C_27_H_25_N_2_O [M + H]^+^: 393.1961, found 393.1959 (−0.5 ppm).

*6-Hexyl-2-(p-tolyl)benzo[gh]perimidine-7-carbaldehyde* (**16b**): This compound was prepared by employing 2-(*p*-tolyl)-1*H*-perimidine **1b** and 5-(hept-1-yn-1-yl)pyrimidine **5a** in a yield of 154 mg (0.38 mmol, 38%). Purification was performed by column chromatography (EtOAc/Hexane = 1:3). The titled compound was obtained as light-brown powder, m.p. 192.1–193.9 °C, R*_f_* 0.62 (EtOAc/Hexane, 1:3). ^1^H NMR (400 MHz, CDCl_3_) *δ* 10.77 (s, 1H), 8.92–8.79 (m, 2H), 8.73 (d, *J* = 7.9 Hz, 2H), 8.54 (d, *J* = 9.2 Hz, 1H), 8.36 (d, *J* = 9.5 Hz, 1H), 8.26 (d, *J* = 9.2 Hz, 1H), 7.41 (d, *J* = 7.9 Hz, 2H), 3.89–3.72 (m, 2H), 2.49 (s, 3H), 1.91–1.78 (m, 2H), 1.66–1.54 (m, 2H), 1.46–1.29 (m, 4H), 0.92 (t, *J* = 6.9 Hz, 3H); ^13^C{^1^H} NMR (101 MHz, CDCl_3_) *δ* 190.8, 161.3, 157.6, 155.3, 149.3, 145.1, 141.9, 137.4, 134.7, 133.6, 130.8, 130.3(2С), 128.4, 127.4(2С), 125.8, 124.7, 123.9, 113.8, 34.4, 31.6, 29.8, 29.7, 28.0, 22.6, 21.9, 14.1; FTIR, *v_max_*: 2960, 2926, 2858, 1954, 1761, 1699, 1687, 1628, 1512, 1408, 1399, 1308, 1179, 957, 854, 830, 725, 669 cm^−1^; HRMS (ESI TOF) *m*/*z*: calc’d for C_28_H_27_N_2_O [M + H]^+^: 407.2118, found 407.2114 (−1.0 ppm).

*2-(4-Ethylphenyl)-6-hexylbenzo[gh]perimidin-7-carbaldehyde* (**16c**): This compound was prepared by employing 2-(4-ethylphenyl)-1*H*-perimidine **1c** and 5-(hept-1-yn-1-yl)pyrimidine **5a** in a yield of 154 mg (0.38 mmol, 38%). Purification was performed by column chromatography (EtOAc/Hexane = 1:3). The titled compound was obtained as light-yellow powder, m.p. 176–177.2 °C, R*_f_* 0.64 (EtOAc/Hexane, 1:3). ^1^H NMR (400 MHz, CDCl_3_) *δ* 10.70 (s, 1H), 8.77–8.68 (m, 4H), 8.43 (d, *J* = 9.2 Hz, 1H), 8.24 (d, *J* = 9.5 Hz, 1H), 8.14 (d, *J* = 9.1 Hz, 1H), 7.42 (d, *J* = 8.0 Hz, 2H), 3.77–3.67 (m, 2H), 2.79 (q, *J* = 7.6 Hz, 2H), 1.81 (p, *J* = 7.7 Hz, 2H), 1.58 (p, *J* = 7.1 Hz, 2H), 1.43–1.30 (m, 7H), 0.92 (t, *J* = 6.9 Hz, 3H); ^13^C{^1^H} NMR (101 MHz, CDCl_3_) *δ* 192.0, 162.8, 155.3, 154.4, 147.7, 144.7, 136.7, 135.8, 132.7, 132.0, 131.0, 129.3(2C), 128.5(2C), 128.2, 128.1, 127.7, 127.6, 125.6, 114.6, 33.8, 31.8, 29.9, 29.1, 27.6, 22.8, 15.6, 14.2; FTIR, *v_max_*: 2963, 2928, 2858, 1918, 1772, 1684, 1626, 1509, 1407, 1396, 1307, 1177, 1017, 957, 844, 687, 668 cm^−1^; HRMS (ESI TOF) *m*/*z*: calc’d for C_29_H_29_N_2_O [M + H]^+^: 421.2274, found 421.2271 (−0.7 ppm).

*6-Hexyl-2-(4-isopropylphenyl)benzo[gh]perimidin-7-carbaldehyde* (**16d**): This compound was prepared by employing 2-(4-isopropylphenyl)-1*H*-perimidine **1d** and 5-(hept-1-yn-1-yl)pyrimidine **5a** in a yield of 134 mg (0.31 mmol, 31%). Purification was performed by column chromatography (EtOAc/Hexane = 1:3). The titled compound was obtained as light-brown powder, m.p. 186.2–187.9 °C, R*_f_* 0.63 (EtOAc/Pe, 1:3). ^1^H NMR (400 MHz, CDCl_3_) *δ* 10.68 (s, 1H), 8.69 (d, *J* = 8.2 Hz, 4H), 8.39 (d, *J* = 9.0 Hz, 1H), 8.19 (d, *J* = 9.4 Hz, 1H), 8.09 (d, *J* = 9.0 Hz, 1H), 7.45 (d, *J* = 8.3 Hz, 2H), 3.69 (t, *J* = 8.2 Hz, 2H), 3.05 (hept, *J* = 6.8 Hz, 1H), 1.79 (p, *J* = 8.0 Hz, 2H), 1.57 (p, *J* = 7.1 Hz, 2H), 1.43–1.30 (m, 10H), 0.92 (t, *J* = 6.9 Hz, 3H); ^13^C{^1^H} NMR (101 MHz, CDCl_3_) *δ* 192.0, 162.8, 155.2, 154.3, 152.3, 144.6, 136.6, 136.1, 132.2, 132.0, 131.0, 129.3(2C), 128.2, 128.1, 127.7, 127.6, 127.0(2C), 125.5, 114.6, 34.3, 33.8, 31.8, 29.9, 27.6, 24.04, 22.8, 14.2(2C); FTIR, *v_max_*: 2957, 2859, 1942, 1774, 1685, 1627, 1508, 1396, 1306, 1017, 912, 849, 690 cm^−1^; HRMS (ESI TOF) *m*/*z*: calc’d for C_30_H_31_N_2_O [M + H]^+^: 435.2431, found 435.2429 (−0.4 ppm).

*6-Hexyl-2-(4-methoxyphenyl)benzo[gh]perimidine-7-carbaldehyde* (**16e**): This compound was prepared by employing 2-(4-methoxyphenyl)-1*H*-perimidine **1e** and 5-(hept-1-yn-1-yl)pyrimidine **5a** in a yield of 148 mg (0.35 mmol, 35%). Purification was performed by column chromatography (EtOAc/Hexane = 1:3). The titled compound was obtained as light-yellow powder, m.p. 178.4–179.7 °C, R*_f_* 0.41 (EtOAc/Hexane, 1:3). ^1^H NMR (400 MHz, CDCl_3_) *δ* 10.77 (s, 1H), 9.10 (d, *J* = 9.5 Hz, 1H), 9.05 (s, 1H), 8.85 (d, *J* = 9.3 Hz, 1H), 8.74 (d, *J* = 9.0 Hz, 2H), 8.33 (d, *J* = 9.5 Hz, 1H) 8.24 (d, *J* = 9.3 Hz, 1H), 7.18 (d, *J* = 9.0 Hz, 2H), 3.90 (s, 3H), 3.86 (t, *J* = 8.1 Hz, 2H), 1.82–1.70 (m, 2H), 1.63–1.49 (m, 2H), 1.43–1.26 (m, 4H), 0.88 (t, *J* = 7.0 Hz, 3H); ^13^C{^1^H} NMR (101 MHz, DMSO-*d*_6_) *δ* 193.2, 161.8, 161.7, 154.9, 154.1, 144.7, 137.4, 137.1, 133.2, 132.1, 131.1, 130.6, 130.3(2С), 127.7, 127.2, 127.1, 124.8, 114.2(2С), 113.9, 55.4, 33.4, 31.1, 28.9, 26.6, 22.1, 14.0; FTIR, *v_max_*: 2958, 2923, 2724, 2048, 1946, 1802, 1696, 1628, 1603, 1510, 1399, 1303, 1249, 1168, 1038, 957, 907, 849, 729 cm^−1^; HRMS (ESI TOF) *m*/*z*: calc’d for C_28_H_27_N_2_O_2_ [M + H]^+^: 423.2068, found 423.2067 (0.1 ppm).

*6-Hexyl-2-(o-tolyl)benzo[gh]perimidin-7-carbaldehyde* (**16f**): This compound was prepared by employing 2-(*o*-tolyl)-1*H*-perimidine **1f** and 5-(hept-1-yn-1-yl)pyrimidine **5a** in a yield of 114 mg (0.28 mmol, 28%). Purification was performed by column chromatography (EtOAc/Hexane = 1:3). The titled compound was obtained as light-yellow powder, m.p. 120.5–121.4 °C, R*_f_* 0.55 (EtOAc/Hexane, 1:3). ^1^H NMR (400 MHz, CDCl_3_) *δ* 10.81 (s, 1H), 8.97–8.86 (m, 2H), 8.59 (d, *J* = 8.6 Hz, 1H), 8.42 (d, *J* = 9.5 Hz, 1H), 8.31 (d, *J* = 9.1 Hz, 1H), 8.07–7.97 (m, 1H), 7.51–7.33 (m, 3H), 3.94–3.75 (m, 2H), 2.66 (s, 3H), 1.84 (m, 2H), 1.60 (p, *J* = 7.3 Hz, 2H), 1.47–1.29 (m, 4H), 0.92 (t, *J* = 6.6 Hz, 3H); ^13^C{^1^H} NMR (101 MHz, CDCl_3_) *δ* 192.0, 165.6, 155.0, 154.1, 145.2(2C), 137.6, 137.4, 133.0, 132.4, 131.5, 131.4, 131.3, 129.8, 128.6(2C), 127.9, 127.5, 126.4, 125.5, 114.3, 34.0, 31.8, 29.8, 27.7, 22.8, 21.1, 14.2; FTIR, *v_max_*: 3353, 3062, 2958, 2855, 1946, 1683, 1628, 1509, 1396, 1308, 1198, 1077, 956, 908, 853, 821, 791, 741, 724, 673 cm^−1^; HRMS (ESI TOF) *m*/*z*: calc’d for C_28_H_27_N_2_O [M + H]^+^: 407.2118, found 407.2112 (1.4 ppm).

*6-Hexyl-2-(2-methoxyphenyl)benzo[gh]perimidine-7-carbaldehyde* (**16g**): This compound was prepared b*y* employing 2-(2-methoxyphenyl)-1*H*-perimidine**1g** and5-(hept-1-yn-1-yl)pyrimidine 5a in a yield of 97 mg (0.23 mmol, 23%). Purification was performed by column chromatography (EtOAc/Hexane = 1:3). The titled compound was obtained as light-red powder, m.p. 138.5–140.6 °C, R*_f_* 0.15 (EtOAc/Hexane, 1:3). ^1^H NMR (400 MHz, CDCl_3_) *δ* 10.76 (s, 1H), 9.12–9.03 (m, 2H), 8.85 (d, *J* = 9.2 Hz, 1H), 8.31 (d, *J* = 9.5 Hz, 1H), 8.21 (d, *J* = 9.2 Hz, 1H), 7.72 (d, *J* = 7.6 Hz, 1H), 7.54 (t, *J* = 7.6 Hz, 1H), 7.24 (d, *J* = 8.4 Hz, 1H), 7.14 (t, *J* = 7.4 Hz, 1H), 3.87–3.79 (m, 5H), 1.73 (brs, 2H), 1.53 (brs, 2H), 1.31 (brs, 4H), 0.87 (t, *J* = 6.9 Hz, 3H); ^13^C{^1^H} NMR (101 MHz, DMSO-*d*_6_) *δ* 193.3, 163.9, 157.4, 154.5, 153.7, 144.6, 137.3, 133.2, 132.3, 131.6, 131.0, 130.7, 129.8, 127.9, 127.6, 127.4, 127.0, 124.5, 120.3, 113.6, 112.3, 55.8, 33.5, 31.2, 29.0, 26.7, 22.2, 14.0; FTIR, *v_max_*: 2925, 2843, 1766, 1690, 1628, 1512, 1248, 1023, 954, 856, 746, 665 cm^−1^; HRMS (ESI TOF) *m*/*z*: calc’d for C_28_H_27_N_2_O_2_ [M + H]^+^: 423.2067, found 423.2067 (0.1 ppm).

*6-(Benzylamino)-2-phenylbenzo[gh]perimidin-7-carbaldehyde* (**17a**): This compound was prepared by employing 2-phenyl-1*H*-perimidine **1a** and 5-(phenylethynyl)pyrimidine **5b** in a yield of 144 mg (0.35 mmol, 35%). Purification was performed by column chromatography (EtOAc/Hexane = 1:3). The titled compound was obtained as gray powder, m.p. 241–242.2 °C, R*_f_* 0.16 (EtOAc/Hexane, 1:1). ^1^H NMR (400 MHz, CDCl_3_) *δ* 8.79 (d, *J* = 6.7 Hz, 2H), 8.63 (d, *J* = 9.4 Hz, 1H), 8.49 (d, *J* = 9.4 Hz, 1H), 8.37 (s, 1H), 8.23 (d, *J* = 9.4 Hz, 1H), 8.09 (d, *J* = 9.4 Hz, 1H), 7.92 (s, 1H), 7.67–7.49 (m, 8H), 6.36 (m, 1H), 5.22 (d, *J* = 5.7 Hz, 2H); ^13^C{^1^H} NMR (101 MHz, CDCl_3_) *δ* 160.9, 154.2, 154.0, 142.3, 139.3, 139.0, 135.0, 134.3, 131.3, 130.8(3C), 129.6, 129.1(2C), 129.0(2C), 128.8(2C), 128.4, 128.0, 127.5, 126.9, 126.6, 123.4, 121.6, 115.0, 39.9; FTIR, *v_max_*: 3304, 3065, 2889, 2714, 1937, 1798, 1732, 1632, 1504, 1412, 1352, 1265, 1207, 899, 851, 835, 775 cm^−1^; HRMS (ESI TOF) *m*/*z*: calc’d for C_28_H_20_N_3_O [M + H]^+^: 414.1601, found 414.1599 (0.4 ppm).

*6-(Benzylamino)-2-(p-tolyl)benzo[gh]perimidine-7-carbaldehyde* (**17b**): This compound was prepared by employing 2-(*p*-tolyl)-1*H*-perimidine **1b** and 5-(phenylethynyl)pyrimidine **5b** in a yield of 196 mg (0.36 mmol, 36%). Purification was performed by column chromatography (EtOAc/Hexane = 1:3). The titled compound was obtained as light-yellow powder, m.p. 268–269 °C, R*_f_* 0.19 (EtOAc/Hexane, 1:1). ^1^H NMR (400 MHz, CDCl_3_) *δ* 9.01 (d, *J* = 9.4 Hz, 1H), 8.87 (s, 1H), 8.68 (d, *J* = 7.9 Hz, 2H), 8.56 (d, *J* = 9.4 Hz, 1H), 8.31 (d, *J* = 9.4 Hz, 1H), 8.26–8.19 (m, 2H), 8.15 (s, 1H), 7.69–7.58 (m, 5H), 7.43 (d, *J* = 7.9 Hz, 2H), 5.19 (d, *J* = 6.0 Hz, 2H), 2.44 (s, 3H); ^13^C{^1^H} NMR (101 MHz, DMSO-*d*_6_) *δ* 161.2, 161.0, 153.6, 153.5, 141.5, 140.6, 138.9, 137.7, 135.6, 134.1, 132.5, 130.5(2С), 129.4(2С), 129.3, 128.8(2С), 128.4(2С), 128.3, 127.0, 126.9, 126.0, 125.5, 122.7, 114.4, 38.5, 21.1; FTIR, *v_max_*: 3285, 3032, 2874, 1919, 1732, 1661, 1557, 1497, 1410, 1391, 1342, 1179, 835, 758 cm^−1^; HRMS (ESI TOF) *m*/*z*: calc’d for C_29_H_22_N_3_O [M + H]^+^: 428.1757, found 428.1759 (0.5 ppm).

*6-(Benzylamino)-2-(4-ethylphenyl)benzo[gh]perimidin-7-carbaldehyde* (**17c**): This compound was prepared by employing 2-(4-ethylphenyl)-1*H*-perimidine **1c** and 5-(phenylethynyl)pyrimidine **5b** in a yield of 141 mg (0.32 mmol, 32%). Purification was performed by column chromatography (EtOAc/Hexane = 1:3). The titled compound was obtained as light-yellow powder, m.p. 233–234.5 °C, R*_f_* 0.19 (EtOAc/Hexane, 1:1). ^1^H NMR (400 MHz, CDCl_3_) *δ* 9.04 (d, *J* = 9.5 Hz, 1H), 8.90 (s, 1H), 8.72 (d, *J* = 8.0 Hz, 2H), 8.59 (d, *J* = 9.5 Hz, 1H), 8.35 (d, *J* = 9.5 Hz, 1H), 8.28–8.23 (m, 2H), 8.17 (s, 1H), 7.73–7.61 (m, 5H), 7.48 (d, *J* = 8.0 Hz, 2H), 5.21 (d, *J* = 6.0 Hz, 2H), 2.75 (q, *J* = 7.5 Hz, 2H), 1.29 (t, *J* = 7.6 Hz, 3H); ^13^C{^1^H} NMR (101 MHz, DMSO-*d*_6_) *δ* 161.2, 161.0, 153.6 (2С), 146.8, 141.6, 138.9, 137.8, 135.8, 134.2, 132.6, 130.5(2C), 129.4, 128.9(2C), 128.5(2C), 128.3, 128.2(2C), 127.1, 127.0, 126.0, 125.5, 122.7, 114.5, 38.5, 28.1, 15.4; FTIR, *v_max_*: 3084, 2968, 1946, 1825, 1776, 1697, 1557, 1495, 1406, 1393, 1339, 1240, 1180, 843 cm^−1^; HRMS (ESI TOF) *m*/*z*: calc’d for C_30_H_24_N_3_O [M + H]^+^: 442.1907, found 442.1916 (−1.6 ppm).

*6-(Benzylamino)-2-(4-isopropylphenyl)benzo[gh]perimidin-7-carbaldehyde* (**17d**): This compound was prepared by employing 2-(4-isopropylphenyl)-1*H*-perimidine **1d** and 5-(phenylethynyl)pyrimidine **5b** in a yield of 169 mg (0.37 mmol, 37%). Purification was performed by column chromatography (EtOAc/Hexane = 1:3). The titled compound was obtained as gray powder, m.p. 246–247.5 °C, R*_f_* 0.19 (EtOAc/Hexane, 1:1). ^1^H NMR (400 MHz, CDCl_3_) *δ* 9.02 (d, *J* = 9.4 Hz, 1H), 8.88 (brs, 1H), 8.71 (d, *J* = 8.2 Hz, 2H), 8.57 (d, *J* = 9.4 Hz, 1H), 8.33 (d, *J* = 9.4 Hz, 1H), 8.27–8.20 (m, 2H), 8.16 (s, 1H), 7.69–7.60 (m, 5H), 7.50 (d, *J* = 8.2 Hz, 2H), 5.20 (d, *J* = 5.9 Hz, 2H), 3.07–2.99 (m, 1H), 1.30 (d, *J* = 7.0 Hz, 6H); ^13^C{^1^H} NMR (101 MHz, DMSO-*d*_6_) *δ* 161.2, 161.0, 153.6 (2C), 151.4, 141.6, 138.9, 137.8, 136.0, 134.1, 132.6, 130.5(2C), 129.4, 128.9(2C), 128.6(2C), 128.3, 127.1, 127.0, 126.8(2C), 126.0, 125.5, 122.7, 114.5, 38.5, 33.5, 23.8 (2C); FTIR, *v_max_*: 3265, 3034, 2949, 2872, 1938, 1738, 1678, 1651, 1557, 1495, 1408, 1385, 1265, 893, 845, 777 cm^−1^; HRMS (ESI TOF) *m*/*z*: calc’d for C_31_H_26_N_3_O [M + H]^+^: 456.2070, found 456.2072 (0.4 ppm).

*6-(benzylamino)-2-(4-methoxyphenyl)benzo[gh]perimidine-7-carbaldehyde* (**17e**): This compound was prepared by employing 2-(4-methoxyphenyl)-1*H*-perimidine **1e** and 5-(phenylethynyl)pyrimidine **5a** in a yield of 124 mg (0.28 mmol, 28%). Purification was performed by column chromatography (EtOAc/Hexane = 1:3). The titled compound was obtained as yellow powder, m.p. 269–270,5 °C, R*_f_* 0.15 (EtOAc/Hexane, 1:1). ^1^H NMR (400 MHz, DMSO-*d*_6_) *δ* 9.02 (d, *J* = 9.4 Hz, 1H), 8.88 (s, 1H), 8.74 (d, *J* = 8.5 Hz, 2H), 8.56 (d, *J* = 9.4 Hz, 1H), 8.32 (d, *J* = 9.4 Hz, 1H), 8.26–8.20 (m, 2H), 8.15 (s, 1H), 7.70–7.60 (m, 5H), 7.17 (d, *J* = 8.5 Hz, 2H), 5.20 (d, *J* = 6.0 Hz, 2H), 3.89 (s, 3H). ^13^C{^1^H} NMR (101 MHz, DMSO-*d*_6_) δ 161.6, 161.2, 160.9, 153.7, 153.6, 141.5, 139.0, 138.9, 137.8, 137.1, 134.1, 132.6, 130.7, 130.5, 130.1(2С), 129.3, 128.8, 128.3, 127.0 (2С), 125.9, 125.4, 125.2, 114.3, 114.2(2С), 55.4, 38.5. FTIR, *v_max_*: 3118, 2831, 2357, 2058, 1926, 1734, 1693, 1602, 1556, 1494, 1408, 1392, 1338, 1253, 1166, 1031, 844, 771 cm^−1^; HRMS (ESI TOF) *m*/*z*: calc’d for C_29_H_22_N_3_O_2_ [M + H]^+^: 444.1705, found 444.1707 (0.4 ppm).

*6-(Benzylamino)-2-(o-tolyl)benzo[gh]perimidine-7-carbaldehyde* (**17f**): This compound was prepared by employing 2-(*o*-tolyl)-1*H*-perimidine 1f and 5-(phenylethynyl)pyrimidine 5b in a yield of 158 mg (0.37 mmol, 37%). Purification was performed by column chromatography (EtOAc/Hexane = 1:3). The titled compound was obtained as light-green powder, m.p. 218–219 °C, R*_f_* 0.19 (EtOAc/Hexane, 1:1). ^1^H NMR (400 MHz, CDCl_3_) *δ* 9.05 (d, *J* = 9.5 Hz, 1H), 8.90 (brs, 1H), 8.60 (d, *J* = 9.5 Hz, 1H), 8.33 (d, *J* = 9.5 Hz, 1H), 8.27–8.17 (m, 3H), 8.07–8.02 (m, 1H), 7.72–7.60 (m, 5H), 7.49–7.35 (m, 3H), 5.22 (d, *J* = 6.0 Hz, 2H), 2.66 (s, 3H); ^13^C{^1^H} NMR (101 MHz, DMSO-*d*_6_) *δ* 164.2, 161.3, 153.2, 153.2, 141.5, 139.1, 138.9, 137.8, 137.1, 134.1, 132.6, 131.2, 131.1, 130.6(2C), 129.4, 129.2, 128.7(2C), 128.3, 127.0, 126.9, 126.1, 125.9, 125.6, 122.5, 113.8, 38.5, 21.1; FTIR, *v_max_*: 3260, 3053, 2889, 2729, 1954, 1829, 1649, 1501, 1410, 1267, 1146, 899, 854, 768 cm^−1^; HRMS (ESI TOF) *m*/*z*: calc’d for C_29_H_22_N_3_O [M + H]^+^: 428.1757, found 428.1762 (−1.0 ppm).

*6-(Benzylamino)-2-(3-hydroxyphenyl)benzo[gh]perimidin-7-carbaldehyde* (**17h**): This compound was prepared by employing 3-(1*H*-perimidin-2-yl)phenol 1h and 5-(phenylethynyl)pyrimidine 5b in a yield of 163 mg (0.38 mmol, 38%). Purification was performed by column chromatography (EtOAc/Hexane = 1:3). The titled compound was obtained as light-yellow powder, m.p. 264–265.5 °C, R*_f_* 0.16 (EtOAc/Hexane, 1:1). ^1^H NMR (400 MHz, CDCl_3_) *δ* 9.69 (s, 1H), 9.01 (d, *J* = 9.5 Hz, 1H), 8.87 (s, 1H), 8.56 (d, *J* = 9.4 Hz, 1H), 8.30 (d, *J* = 9.5 Hz, 1H), 8.27–8.19 (m, 4H), 8.16 (s, 1H), 7.76–7.56 (m, 5H), 7.42 (t, *J* = 8.0 Hz, 1H), 6.98 (d, *J* = 8.0 Hz, 1H), 5.19 (d, *J* = 5.9 Hz, 2H); ^13^C{^1^H} NMR (101 MHz, DMSO-*d*_6_) *δ* 161.7, 161.3, 158.3, 154.0, 153.9, 142.0, 140.1, 139.3, 138.2, 134.6, 133.0, 131.0(2С), 130.2, 129.9, 129.3(2С), 128.7, 127.5, 127.4, 126.5, 126.0, 123.1, 119.8, 118.3, 115.6, 115.0, 39.0; FTIR, *v_max_*: 3374, 3273, 3053, 2864, 2741, 1933, 1792, 1680, 1622, 1582, 1558, 1516, 1410, 1385, 1271, 1238, 1215, 1148, 1078, 1030, 995, 966, 887, 849, 835, 820, 797, 762 cm^−1^; HRMS (ESI TOF) *m*/*z*: calc’d for C_28_H_20_N_3_O_2_ [M + H]^+^: 430.1550, found 430.1541 (−2.1 ppm).


**Isolation of intermediate 1,6,10,10a-tetrahydroquinazolino[6,7,8-*gh*]perimidines 14**


A round bottomed 10 mL flask was charged with 5-(phenylethynyl)pyrimidine (**5b**, 1.00 mmol) and 1*H*-perimidine (**1c** or **1e**, 1.00 mmol) dissolved in methanesulfonic acid was added in one portion. The mixture was stirred at room temperature, and the reaction progress was closely monitored by TLC (eluent ethyl acetate/petroleum ether, 1:3). After about 20 min, the mixture was poured into cold water and neutralized with aqueous ammonia. The formed precipitate was filtered and washed with small portions of water to remove excess ammonia. The crude material was fractionated by preparative column chromatography to isolate compounds **14c** or **14e** in low yields. Attempts to isolate the analogous pentacyclic products from reactions with 5-(heptynyl)pyrimidine (**5a**) were not successful.

*6-benzylidene-2-(4-methoxyphenyl)-6,10,10a,10b-tetrahydroquinazolino[6,7,8-gh]perimidine* (**14c**): The titled compound was obtained as orange powder, m.p. 254.5–256.1 °C, R*_f_* 0.33 (EtOAc/Hexane, 1:1). Yield 40.7 mg (0.09 mmol, 9%). ^1^H NMR (400 MHz, DMSO-*d*_6_) *δ* 8.78 (s, 1H), 8.43 (d, *J* = 8.4 Hz, 1H), 8.11 (s, 1H), 8.04 (d, *J* = 8.2 Hz, 2H), 7.43 (d, *J* = 8.4 Hz, 2H), 7.29 (s, 1H), 7.03 (s, 2H), 6.99–6.92 (m, 2H), 6.90–6.83 (m, 1H), 6.76 (s, 2H), 4.95 (s, 1H), 3.48–3.43 (m, 1H), 3.21–3.11 (m, 1H), 2.71 (q, *J* = 7.5 Hz, 2H), 1.23 (t, *J* = 7.6 Hz, 4H). ^13^C{^1^H} NMR (101 MHz, DMSO-*d*_6_) δ 163.0, 157.1, 156.2, 152.7, 148.2, 144.3, 136.5, 132.8, 131.5, 130.3, 128.4(3С), 127.7(3С), 126.1, 124.7, 122.9, 51.7, 35.9(2С), 28.6, 15.9. FTIR, *v_max_*: 3260, 2962, 2353, 1733, 1552, 1397, 1268, 1268, 1120, 1043, 824, 784, 665 cm^−1^; HRMS (ESI TOF) *m*/*z*: calc’d for C_31_H_25_N_4_ [M + H]^+^: 453.2074, found 453.2073 (−0.2 ppm).

*6-benzylidene-2-(4-methoxyphenyl)-6,10,10a,10b-tetrahydroquinazolino[6,7,8-gh]perimidine* (**14e**): The titled compound was obtained as orange powder, m.p. 220.9–223.7 °C, R*_f_* 0.23 (EtOAc/Hexane, 1:1). Yield 54.5 mg (0.12 mmol, 12%). ^1^H NMR (400 MHz, DMSO-*d*_6_) *δ* 8.76 (s, 1H), 8.41 (d, *J* = 8.4 Hz, 1H), 8.11–8.05 (m, 3H), 7.26 (s, 1H), 7.12 (d, *J* = 8.9 Hz, 2H), 7.05–6.98 (m, 2H), 6.99–6.91 (m, 2H), 6.84 (s, 1H), 6.78–6.70 (m, 2H), 4.93 (s, 1H), 3.85 (s, 3H), 3.41–3.37 (m, 1H), 3.17–3.11 (m, 1H). ^13^C{^1^H} NMR (101 MHz, DMSO) δ 163.1, 162.3, 157.0, 156.1, 153.1, 151.4, 150.1, 145.2, 144.3, 136.9, 136.2, 132.8, 129.3, 128.5(2С), 127.7, 126.1, 125.2, 124.5, 124.0, 122.6, 115.1(2С), 114.4, 104.7, 104.0, 55.9, 51.7, 35.9(2С). FTIR, *v_max_*: 3201, 2357, 1735, 1564, 1395, 1258, 1174, 1045, 834, 647 cm^−1^; HRMS (ESI TOF) *m*/*z*: calc’d for C_30_H_23_N_4_O [M + H]^+^: 455.1866, found 455.1867 (0.1 ppm).

## 4. Conclusions

Unusual Brønsted acid-catalyzed cascade transformations were shown to accompany the peri*-*annulation reaction of perimidines in the presence of 5-alkynylpyrimidines. The reactivity pattern varies depending on the nature of the acetylene substrate. Reactions involving alkylacetylene **5a** produced 6-alkyl-7-formyl-1,3-diazopyrenes **16**, which was rationalized by the acid-assisted hydrolysis of the initially formed peri-annulation product, dihydroquinazolino[6,7,8-*gh*]perimidine **15**. Peri-Annulation employing arylacetylene **5b** took a different route involving a 1,2-benzyl shift, to afford 6-benzylamino-7-formyl-1,3-diazopyrene **17**. Despite modest chemical yields, the presented method has a significant practical value, as it allows for a rapid increase of molecular complexity and provides easy access to useful heterocyclic synthons with conveniently placed functional handles.

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
