# Peer review of "Annulation of Perimidines with 5-Alkynylpyrimidines en Route to 7-Formyl-1,3-Diazopyrenes"

_ijms, 2022, doi:10.3390/ijms232415657_

Round 1
Reviewer 1 Report
The manuscript describes an unusual Bronsted acid-catalyzed cascade transformations for annulation of perimidines with 5-alkynylpyrimidines to obtain substituted 7-formyl-1,3-diazopyrene. The reactivity pattern of the reaction depends on the nature of the alkyne used. This method provides an easy access to useful complex heterocyclic scaffold and allows for the variation of the substituents. Despite its low-yielding feature, this process would be interesting to the synthetic community working on materials science and drug design. For this reason, this reviewer supports publication of the manuscript.
Author Response
Reviewer 1
Comments and Suggestions for Authors
The manuscript describes an unusual Bronsted acid-catalyzed cascade transformations for annulation of perimidines with 5-alkynylpyrimidines to obtain substituted 7-formyl-1,3-diazopyrene. The reactivity pattern of the reaction depends on the nature of the alkyne used. This method provides an easy access to useful complex heterocyclic scaffold and allows for the variation of the substituents. Despite its low-yielding feature, this process would be interesting to the synthetic community working on materials science and drug design. For this reason, this reviewer supports publication of the manuscript.
Response: We are grateful to this reviewer for their supportive comments.
Reviewer 2 Report
The paper has shown quite an interesting reaction found during their studies. In order to understand the reaction, they tried to come up with some explanations. However, the paper's mechanisms are all based on surmises. No evidence and no designed experiments to prove their proposed mechanisms, which is unacceptable. Some of the experiments are not hard to design.
1. For scheme 1, please have the citation discussed in the legend. For example, organize the content into (A), (B), and (C) sections, which have (A) Borovlev's work, (B) Aksenov's work, and (C) this work. In the legend, should also describe the main points briefly and have the citation.
2. Please include the yield in each scheme. That should be common sense in organic chemistry manuscripts to report the yields and conditions for each reaction.
3. Please flip some of the molecular structures. For example, compound 16 and compound 15 should have the R substitute group in the same position for a better understanding. It happened in Scheme 3, Scheme 4, and Scheme 5.
4. The yield of each reaction is low, which suggests multiple reactions were going on. Do you have any other products that get separated? How do you confirm your product? Only NMR and mass spec are not enough for this compound. Did you collect crystal structures? How did that go?
5. A proposed mechanism needs some evidence to confirm. The benzyl group sure has more 1,2 migration trend in this case; why did you not try to design a different compound with a similar 1,2 migration tendency or visa versa?
6. Scheme 4 has repeated information.
Author Response
Reviewer 2
Comments and Suggestions for Authors
The paper has shown quite an interesting reaction found during their studies. In order to understand the reaction, they tried to come up with some explanations. However, the paper's mechanisms are all based on surmises. No evidence and no designed experiments to prove their proposed mechanisms, which is unacceptable. Some of the experiments are not hard to design.
- For scheme 1, please have the citation discussed in the legend. For example, organize the content into (A), (B), and (C) sections, which have (A) Borovlev's work, (B) Aksenov's work, and (C) this work. In the legend, should also describe the main points briefly and have the citation.
Response: The scheme was modified as suggested. We do not feel placing the extended description in the legends is necessary, since it will make a scheme heavy and non-readable. Furthermore, all the details are provided in the related text.
- Please include the yield in each scheme. That should be common sense in organic chemistry manuscripts to report the yields and conditions for each reaction.
Response: Providing yields in each scheme is not possible. Some of the schemes outline hypothetical processes (method design), some other – mechanistic rationales. Yields are shown in schemes describing preparative experiments in the frame of the presented methodology.
- Please flip some of the molecular structures. For example, compound 16 and compound 15 should have the R substitute group in the same position for a better understanding. It happened in Scheme 3, Scheme 4, and Scheme 5.
Response: The described problem occurred in Scheme 4 only. This scheme was modified as suggested. The rest of the Schemes were perfectly fine from this prospective. Most likely, the depicted ANRORC process involving disconnection of the bond and reconnection in other position is harder for understanding, but it has nothing to do with orientation of the molecules.
- The yield of each reaction is low, which suggests multiple reactions were going on. Do you have any other products that get separated?
Response: This question was already fully addressed in the original text of the manuscript (in first and second paragraphs on page 4). Compounds 16 and 17 were the only isolable products. The rest of materials turned into heavy tar, that did not move through a chromatographic column.
How do you confirm your product? Only NMR and mass spec are not enough for this compound. Did you collectcrystal structures? How did that go?
Response: We did not succeed in growing crystal suitable for X-ray diffraction study. However, the structure of compound 17 was unambiguously confirmed using 2D NMR spectroscopy methods. We added these materials tothe Supporting Info.
5. A proposed mechanism needs some evidence to confirm. The benzyl group sure has more 1,2 migration trend in this case; why did you not try to design a different compound with a similar 1,2 migration tendency or visa versa?
Response: We were able to isolate intermediate 1,6,10,10a-tetrahydroquinazolino[6,7,8-gh]perimidines 14c,e in low yields after quenching the corresponding reactions after 20 min (Scheme 3). These stable crystalline substances were purified and re-subjected back to the same reaction conditions to afford the expected products 17c,e, thus confirming the originally proposed rationale. Probably, alternative pathways for migration suggested by reviewer do exist, but most likely, they lead to the formation of less stable products, which decompose or polymerize. Design of alternative cascade transformations based on different migrations of the benzylic groups is very intriguing task. It should be pointed out, however, that such task clearly exceeds the goals of the featured studies and could be worthy of a separate report.
Reviewer 3 Report
In this manuscript, the authors describe an unusual rearrangement when reacting perimidines with alkynylpyrimidyne. Depending on the nature of the group on the alkyne, differently functionalized 1,3-diazopyrenes (with aldehydes or with aldehydes plus amines) are obtained after peri-annulation.
The manuscript is well written and the molecules described are pretty well characterized. Although the yields are modest at best, the mechanisms involved in the synthesis are interesting cascade transformations that enable the preparation of functionalized diazopyrene (that can be further modified for potential applications).
I think that the results presented in this article can attract the interest of the readers of International Journal of Molecular Sciences, and deserve publication after some minor corrections:
- line 41: compounds 4 are described as triazines, but in scheme 1, 4 seem to be pyrimidines. Is there any mistake in the text or the scheme?
- Scheme 4: there seems to be some problem with this scheme, the part in page 4 is a duplicate of page 3.
Author Response
Reviewer 3
Comments and Suggestions for Authors
In this manuscript, the authors describe an unusual rearrangement when reacting perimidines with alkynylpyrimidyne. Depending on the nature of the group on the alkyne, differently functionalized 1,3-diazopyrenes (with aldehydes or with aldehydes plus amines) are obtained after peri-annulation.
The manuscript is well written and the molecules described are pretty well characterized. Although the yields are modest at best, the mechanisms involved in the synthesis are interesting cascade transformations that enable the preparation of functionalized diazopyrene (that can be further modified for potential applications).
I think that the results presented in this article can attract the interest of the readers of International Journal of Molecular Sciences, and deserve publication after some minor corrections:
- line 41: compounds 4 are described as triazines, but in scheme 1, 4 seem to be pyrimidines. Is there any mistake in the text or the scheme?
Response: We are grateful for catching this mistake. Of course, these are pyrimidines, the text was corrected.
- Scheme 4: there seems to be some problem with this scheme, the part in page 4 is a duplicate of page 3.
Response: We do not see any problems with this scheme in our copy of the files. It shows only in page 3, as one copy. Most likely, this was an artefact of cross-platform conversion of the files?
Round 2
Reviewer 2 Report
Thank you for the answers. The 2D NMR spectra are good to demonstrate the configuration of these aromatic molecules. With these revision, it is much better than the previous version.
(1) Are figure S2 and figure S4 13C DEPTQ spectra? What type of DEPTQ were you running for these molecules? Was it DEPTQ-135?
(2) Please also show the nitrogen region in the HSQC and HMBCGP in figure S52 and figure S53.
Author Response
We are grateful to this reviewer for his patient and meticulous work helping us to improve the quality of our text. We did our best to address all the indicated issues.
1) Indeed, DEPTQ-135 experiments were used to measure 13C NMR chemical shits. This is shown now in legends for all relevant spectral charts.
2) As requested, 1D projections were added to 15N axis of 1H-15N HSQC and 1H-15N HMBCGP spectra (we do not have independently recorded 15N NMR spectrum, as it would be very hard to measure this using non-enriched sample).
3) The text was carefully proof-read by a native speaker holding PhD degree in organic chemistry, which helped to improve the text quality and eliminate all possible spelling mistake and grammar issues.